# Melittin from Bee Venom Encapsulating Electrospun Fibers as a Potential Antimicrobial Wound Dressing Patches for Skin Infections

**DOI:** 10.3390/pharmaceutics14040725

**Published:** 2022-03-28

**Authors:** Walaa S. Aburayan, Areej M. Alajmi, Ahmed J. Alfahad, Wijdan K. Alsharif, Abdullah A. Alshehri, Rayan Y. Booq, Samar A. Alsudir, Fatemah M. Alsulaihem, Haitham A. Bukhary, Moutaz Y. Badr, Essam J. Alyamani, Essam A. Tawfik

**Affiliations:** 1National Center of Biotechnology, Life Science and Environment Research Institute, King Abdulaziz City for Science and Technology (KACST), Riyadh 11442, Saudi Arabia; waburayan@kacst.edu.sa (W.S.A.); amaalajmi@kacst.edu.sa (A.M.A.); ajlfahad@kacst.edu.sa (A.J.A.); walsharif@kacst.edu.sa (W.K.A.); abdualshehri@kacst.edu.sa (A.A.A.); rbooq@kacst.edu.sa (R.Y.B.); salsadeer@kacst.edu.sa (S.A.A.); falsulaihem@kacst.edu.sa (F.M.A.); eyamani@kacst.edu.sa (E.J.A.); 2Department of Pharmaceutics, College of Pharmacy, Umm Al-Qura University, Makkah 24381, Saudi Arabia; habukhary@uqu.edu.sa (H.A.B.); mybadr@uqu.edu.sa (M.Y.B.)

**Keywords:** electrospun fibers, electrospinning, bee venom, melittin, skin infection, antimicrobial wound dressing, antimicrobial resistant bacteria

## Abstract

Skin infection compromises the body’s natural defenses. Several antibiotics are no longer effective owing to the evolution of antimicrobial-resistant (AMR) bacteria, hence, the constant development of novel antibacterial agents. Naturally occurring antibacterial agents may be potential candidates for AMR bacterial infection treatments; however, caution should be taken when administering such agents due to the high incidence of toxicity. A fibrous material system from a biocompatible polymer that could be used as a skin patch for skin infections treatment caused by AMR bacteria is proposed in this study. Bee venom’s active ingredient, melittin, was fabricated using electrospinning technology. Scanning electron microscopy showed that melittin-loaded fibers had smooth surfaces with no signs of beads or pores. The average diameter of this fibrous system was measured to be 1030 ± 160 nm, indicating its successful preparation. The melittin fibers’ drug loading and entrapment efficiency (EE%) were 49 ± 3 µg/mg and 84 ± 5%, respectively. This high EE% can be another successful preparatory criterion. An in vitro release study demonstrated that 40% of melittin was released after 5 min and achieved complete release after 120 min owing to the hydrophilic nature of the PVP polymer. A concentration of ≤10 µg/mL was shown to be safe for use on human dermal fibroblasts HFF-1 after 24-h exposure, while an antibacterial MIC study found that 5 μg/mL was the effective antimicrobial concentration for *S. aureus*, *A. baumannii*, *E. coli* and *Candida albicans* yeast. A melittin-loaded fibrous system demonstrated an antibacterial zone of inhibition equivalent to the control (melittin discs), suggesting its potential use as a wound dressing patch for skin infections.

## 1. Introduction

Although certain bacterial strains may develop antibacterial resistance, almost all skin infections are clinically treated using antibiotics [1]. Adverse events associated with the use of antibiotics can range from mild to severe, depending on the type of antibiotic being used. These side effects may include digestive problems, such as nausea, vomiting, diarrhea, bloating, a sensation of fullness, stomach cramps or discomfort. Antibiotics are intended to prevent and inhibit bacteria that are dangerous to humans; however, they can also eliminate the beneficial ones that protect humans from other illnesses, such as fungal infections [1]. Previous studies have shown that naturally occurring antimicrobial peptides (AMPs) can overcome bacterial resistance owing to their ability to penetrate the cell membranes of targeted bacteria, and are less likely to be rejected by bacterial genetic factors [2,3,4,5]. The discovery of high-throughput AMP sources have received significant attention over the last three decades. Nevertheless, most AMPs exhibit strong membrane binding and a safety issue against mammalian cells, limiting their potential use as therapeutic agents [6,7,8].

Nanotechnology has been utilized to improve the AMP’s targeted delivery to the infected tissues to circumvent their inherent cytotoxicity [9,10]. In contrast to low-molecular-weight antibiotics, the entrapment of AMPs into nano-based materials and the control of their release might be difficult owing to the instability of AMPs and the amphiphilicity of secondary structures [11,12,13]. Complex processes are often required for the preparation of multicomponent nano-formulations. As a result, it is critically important to create simple but effective methods that enable AMPs to be integrated into a nano-based material whilst retaining their antibacterial selectivity and efficacy. This approach could significantly improve AMPs therapeutic index, thus increasing their potential application as an effective antimicrobial agent in clinical settings.

To successfully interface with tissues and cells in a predictable and regulated manner, electrospun nanofibers provide a unique platform. They regulate the molecular packing and internal order of therapies, specifying critical criteria for a practical interface with cells and tissues. Incorporating various functional peptides or proteins with a specific biocompatible polymer may result in highly modular molecular building blocks that can be used in many different applications. The assembly of these fibers allows the fabrication of nanofibers displaying multivalent ligands and epitopes, which could be tailored to have precisely regulated multivalency, density, stoichiometry and recognition sites, to perform a desired biological function. Consequently, angiogenic nanofibers [14,15], supramolecular nanofiber vaccines [16,17,18] and fibrous tissue scaffolds [19,20] have all been successfully utilized.

In this study, a nanofibrous scaffold made of polyvinylpyrrolidone (PVP) polymer, a non-ionic, biocompatible and biodegradable polymer that is known to be “generally regarded as safe” for humans, was used to encapsulate a natural AMP, i.e., melittin from bee venom (BV) through electrospinning. This wound dressing scaffold is intended to apply to antimicrobial-resistant (AMR) bacteria-infected skin. Electrospinning is a technique that creates long fibrous-like structures with diameters in the nano-scale via an electric field on a viscous polymer solution, applied in several pharmaceutical applications (i.e., wound healing) [21]. Additionally, electrospun nanofibers are also favorable for biomedical application owing to their high surface area and pore size, both of which are beneficial for cell development, and their resemblance with extracellular matrix (ECM) proteins that are an essential factor in the wound healing process [21].

Melittin peptide is the most abundant component of BV (40–48%, *w*/*w*) and possesses a strong cytolytic and antimicrobial effect against a broad spectrum of bacterial strains [22]. Melittin peptide is highly effective when given provided it is done so with optimal timing and concentration [23]. Antibacterial drugs often target the cell membrane essential for bacterial survival/growth by separating energy balances and intracellular materials. Therefore, an antibacterial agent’s efficacy depends on whether it can damage the cell membrane structure or interfere with an enzyme system’s ability to suppress bacterial growth [24]. It was previously reported that BV could destroy the integrity of the bacterial outer membrane, creating pores that enable membrane permeability which is critical for bacterial recovery and survival [25,26]. However, lipopolysaccharides (LPS) in gram-negative bacteria’s outer membrane prevent melittin from penetrating the cytoplasmic membrane [27]. Phospholipid hydrolysis occurs modestly over an extended period, disrupting the gram-negative bacteria cell membrane [27]. The antibacterial mechanisms of action of melittin could not be linked to BV in gram-negative bacteria [28].

Although BV’s mode of action against bacteria is not yet completely understood, the findings of the current work established that the active compound of BV, melittin, fibers might be utilized in pharmaceutical applications as a potential natural antibacterial wound dressing for skin infections. Furthermore, BV was previously reported to be used for wound healing due to it promoting skin cell regeneration [29]. This can be an additional advantage in treating skin infections, i.e., inhibiting AMR bacteria and promoting wound healing.

## 2. Materials and Methods

### 2.1. Materials

Polyvinylpyrrolidone (PVP) was obtained from Sigma–Aldrich (St. Louis, MO, USA). The average molecular weight of PVP was 1,300,000 dalton. Melittin (honeybee venom, 98.02%) was purchased from Ontores Biotechnologies (Hangzhou, Zhejiang, China). Acetonitrile solvent was bought from PanReac AppliChem ITW Reagents (Barcelona, Spain), whereas ethanol (≥99.5%), formic acid, KCl (99.0–100.5%), KH₂PO₄ (≥99.0%), NaCl (≥99.5%) and Na₂HPO₄ (≥ 99.0%) were all obtained from Sigma–Aldrich (St. Louis, MO, USA). Milli Q, Millipore was used to generate distilled water (Billerica, MA, USA).

### 2.2. Preparation of Fibers Using Electrospinning

PVP in a concentration of 8% *w*/*v* was dissolved initially in absolute ethanol and then stirred for 2 h at room temperature. With a concentration of 0.5% (*w*/*v*), melittin was mixed with the solution of PVP in a polymer:drug ratio of 16:1, then stirred for an additional hour to achieve a homogenous polymer–drug solution. Melittin-loaded fibers were prepared using the electrospinning setup of Spraybase^®^ (Dublin, Ireland). The flow rate of the spinning solution was controlled at a rate of 1 mL/h using a syringe pump, and a stable jet was obtained at a needle tip and collector distance of 15 cm, 0.9 mm inner needle diameter and a voltage of 9 kV. The electrospinning process was conducted at a relative humidity of 30 to 35% and an ambient temperature, while aluminum foil covering the metal collector was used as a substrate for the deposition of fibers. On the other hand, the blank fibers were prepared under the same conditions but with no melittin added. This method was modified from Aburayan et al. [30].

### 2.3. Melittin-Loaded Fibers Morphology and Diameter Assessment

Scanning electron microscopy (SEM; JSM-IT500HR SEM, JEOL Inc., Peabody, MA, USA) was used to demonstrate the prepared fibers surface morphology and measure their diameter. Fibers collected on aluminum foil were observed under the SEM at an accelerating voltage of 5 kV. The diameter was measured using the SEM software (SEM Operation, 3.010, Akishima, Tokyo, Japan). The average of 20 fibers was measured.

### 2.4. Fourier Transform Infrared (FTIR) of Melittin-Loaded Fibers

FTIR spectra were collected with an Agilent Cary 630 ATR-FTIR analyzer (Agilent Technologies Inc., Santa Clara, CA, USA). Each sample was examined over the range of 4000 to 650 cm^−1^, at 1 cm^−1^ spectral resolution and 4 scans/sample. Melittin’s chemical structure was drawn by chemicalbook.com and shown in Figure 1.

### 2.5. Melittin Quantification Using Ultra-High-Performance Liquid Chromatography-–MS/MS (UHPLC–MS/MS)

The quantification of melittin was performed using a modified method of Huang et al. [31] in a 1290 Infinity II UHPLC system (Agilent, Santa Clara, CA, USA). The UHPLC system consisted of a binary solvent gradient pump with an integrated degasser (G7116A), autosampler (G7167B), column compartments (G7116B), diode array detector (G7117B) and an Agilent Poroshell 120 EC-C_18_ column (4.6 mm × 100 mm, 2.7 μm). The solvents used were LC/MS grade, with a mobile phase of two solutions: Solution A (i.e., water with 0.1% formic acid) and Solution B (i.e., acetonitrile with 0.1% formic acid). The gradient elution method was as follows: 0.0–2.0 min at 5% B, 2.1–9.0 min at 5–95% B, 9.1–12.0 min 95% B, 12.1–13.0 min to switch back to 5% B and finally 13.1–15.0 min to hold at 5% B, with an injection volume of 5 µL, a flow rate of 0.3 mL/minute at 40 °C and a total run time of 15 min.

The identification and quantification of melittin were made on a 6470 Triple Quadrupole mass spectrometer coupled to an electrospray ionization (ESI) interface (Agilent, USA). The mass spectrometer was operated in the positive ion electrospray ionization (ES+), and the electrospray parameters (H-ESI) were as follows: vaporizer temperature 290 °C, sheath gas flow 11.0 L/min, spray voltage 3.0 kV (positive mode) and nebulizer 20 psi. The peak of melittin detection was first based on DAD and MS full scan mode at a retention time (*Rt*) of 7.04 min. The ions *m/z* 712.5 ([M+4H]^4+^), *m*/*z* 570.2 ([M+5H]^5+^) and *m*/*z* 475.3 ([M+6H]^6+^) were characteristic for melittin and present in high abundance. The quintuple-charged molecular ion 570.2 ([M+5H]^5+^) was selected as a precursor ion for quantification since it was the most abundant ion and the MRM transition was (*m*/*z* 570.2→85.9, 45 V). The UHPLC chromatograms of bee venom can be demonstrated in the Appendix A. A standard calibration curve of melittin concentrations, ranging from 200 to 6.25 μg/mL, was plotted using OriginPro 2016 software (OriginLab Corporation, Northampton, MA, USA). All data acquired of melittin quantification was processed using MassHunter software (Agilent, Santa Clara, CA, USA).

### 2.6. Drug Loading (DL) and Entrapment Efficiency (EE%) of the Melittin-Loaded Fibers Determination

The DL and EE% of the melittin fibers were determined by weighing three pieces (16 ± 1 mg) of the fibrous mat and dissolving them in 10 mL distilled water (pH 5.5) in a dark room and at ambient temperature for at least 4 h. Melittin concentrations and amount were analyzed using the previously mentioned UHPLC–MS/MS method, and the DL and EE% were determined according to the following equations:$$DL=\frac{Entrapped\;drug\;amount}{Yield\;of\;fibres\;amount}$$
$$EE=\frac{Actual\;drug\;amount}{Theoretical\;drug\;amount}\times 100$$

The results represent independent triplicates’ average ± standard deviation (SD).

### 2.7. Drug Release of the Melittin-Loaded Fibers

The melittin release was determined by initially weighing three pieces (24 ± 1 mg) of the fibrous mat and dissolving them in 20 mL phosphate buffer saline (PBS; pH 5.5). The PBS was prepared by dissolving 200 mg with adjusted pH to 5.5, which, in by mixing 200 mg KCl, 240 mg KH₂PO₄, 8000 mg NaCl and 1440 mg Na₂HPO₄ in 1 L distilled water, and 5M HCl adjusted the pH to 5.5 to represent the pH of human skin. The release study was conducted using a thermostatic shaking incubator at 37 °C and 50 rpm in a dark room, where 1 mL of samples were withdrawn at a time-point ranging from 5 to 240 min. At the same time, equivalent volumes of pre-warmed fresh buffer were replaced to maintain a sink condition. This method was adapted from Aburayan et al. [30]. The melittin amount was then analyzed using the previously mentioned UHPLC–MS/MS method. The percentage of cumulative release was calculated as a function of time, as follow:$$\mathrm{The}\;\mathrm{cumulative}\;\mathrm{amount}\;\mathrm{of}\;\mathrm{release}\;\left(\%\right)=\frac{\mathrm{Ct}}{C\infty }\times 100$$
where Ct is the melittin amount released at time t, while C∞ is the initial melittin amount. The results represent the average ± SD of independent triplicates. Data were examined by OriginPro 2016 software (OriginLab Corporation, Northampton, MA, USA).

### 2.8. In Vitro Cytotoxicity Assessment of Melittin

The cytotoxicity screening of melittin is an essential step toward biomedical application. In this study, increased compound concentrations were examined in vitro against a human skin fibroblast cell line (HFF-1, ATCC number SCRC-1041) to determine the optimal and safe concentrations used on a living tissue after 24-, 48- and 72-h treatment. The culturing of both cell lines was routinely maintained in DMEM (i.e., Dulbecco’s Modified Eagle’s Medium). The cytotoxicity of the melittin was assessed by measuring the cellular metabolic activity using a MTS assay, following a modified method of Alkahtani et al. [32]. The MTS Reagent (cell Titer 96^®^Aqueous one solution cell proliferation assay) was supplied by Promega (Southampton, UK).

The HFF-1 cells were harvested and counted using trypsin and trypan blue exclusion tests; 1.5 × 10^4^ cells were seeded into each well of the 96-well plate. Cells were then incubated at 5% CO_2_ and 37 °C in a cell culture incubator overnight. 100 µL of increasing melittin concentrations (0.625 to 80 µg/mL) were exposed to the cell lines for 24, 48 and 72 h, while cells were exposed to 0.1% triton x-100 and DMEM only were used as negative and positive controls, respectively. Subsequently, consumed media were aspirated from all wells, and then 100 µL of fresh DMEM were applied, followed by 20 µL of the MTS reagent. The cells were returned to the incubator for another 3 h. The absorbance of MTS solution was measured using Cytation^TM^3 absorbance microplate reader (BIOTEK instruments inc, Winooski, VT, USA) at 490 nm, to measure the cell viability percentage using the following equation:$$\mathrm{Cell}\;\mathrm{Viability}\;\left(\%\right)=\frac{\left(S-T\right)}{\left(H-T\right)}\times 100$$
where S is the absorbance of cells exposed to the tested compound, T is the absorbance of cells exposed to triton x-100 (negative control) and H is the absorbance of cells exposed to DMEM (positive control). The results represent the average ± SD of independent triplicates.

### 2.9. Bacterial Inoculums and Culture Medium Preparation

Five reference strains, from ATCC, of gram-negative and -positive bacteria that included *Pseudomonas aeruginosa* (*P. aeruginosa*) ATCC 27853, *Escherichia coli* (*E. coli*) ATCC 25922, *Acinetobacter baumannii* (*A. baumannii*) ATCC BAA 747, *Staphylococcus aureus* (*S. aureus*) ATCC 976, ATCC 977, ATCC 29,213 and one yeast *Candida albicans* (*C. albicans*) ATCC 66,027 were tested. Additionally, four multi-drug resistant strains were also assessed, including *methicillin-resistant S. aureus* (*MRSA*) ATCC 43300, clinical isolates of *E. coli* MDR 1060, *P. aeruginosa* MDR 7067 and *A. baumannii* MDR 3087.

Bacterial and yeast suspensions of 10^8^ colony-forming units/mL (CFU/mL) were prepared by inoculating colonies in 0.5 McFarland of Mueller–Hinton broth. Bacterial and yeast strains were then cultured on Mueller–Hinton agar and incubated overnight at 37 °C, except for *C. albicans*, which was incubated at 30 °C.

### 2.10. Minimum Inhibitory Concentration (MIC) of Melittin

The MIC for melittin was carried out according to a modified method of Aburayan et al. and Clinical and Laboratory Standards Institute (CLSI) [30,33]. Melittin dissolved in sterile distilled water, in a 40 mg/mL concentration and was serially diluted in 96-well plates containing culture medium at a volume of 100 μL. Then, 100 μL of bacterial or yeast inoculums were added to obtain melittin concentrations ranging from 0.01 to 20 μg/mL, where wells containing bacteria or yeast only were used as growth control. The microplates were incubated at 37 °C, except for the yeast, incubated overnight at 30 °C. MIC is determined as the lowest concentration of which there is no bacterial/yeast growth (i.e., clear well). Readings were measured by a Cytation^TM^ 3 absorbance microplate reader (BIOTEK instruments Inc., Winooski, VT, USA) at 620 nm, and the results represent the average ± SD of independent triplicates. The microbial growth was observed visually by increasing the turbidity of the medium.

### 2.11. Zone of Inhibition Assessment of Melittin-Loaded Fibers

The antimicrobial activity of melittin-loaded fibers was assessed by measuring the inhibition zones against all bacterial and yeast strains that were previously used in the MIC assay (Section 2.9), according to Aburayan et al. [30]. 1 × 10^8^ CFU/mL inoculums were distributed on the agar plates. Approximately 0.5 mg melittin-loaded (≈25 μg of melittin) and blank (melittin-free) fibers were placed on the surface of the agar plates. Melittin discs containing an equal amount of melittin-loaded fibers (≈25 μg of melittin) were used as an experimental control. All agar plates were overnight incubated at 37 °C, except for the yeast, which was incubated at 30 °C. The diameters of the no-growth areas were measured in millimeters (mm).

## 3. Results and Discussion

### 3.1. Fibers’ Morphology and Diameter Analysis

Antimicrobial electrospun fibers applied to the skin of infected wounds holds high potential in the wound healing field due to their ability to topically deliver the antimicrobial agent, preventing systemic drug-associated adverse events and enhancing the patient compliance with comprehensive wound care [34,35,36]. The parameters of the electrospinning process, such as the concentration of polymer used, the flow rate used, the collecting distance and finally, the applied voltage, were optimized to obtain successful fibrous systems. As shown in Figure 2, these fibers demonstrated non-porous and non-beaded surfaces. They were smooth, with equivalent average diameters of 990 ± 130 nm for the blank fibers and 1030 ± 160 nm for the melittin-loaded fibers, indicating successful fiber preparation criteria [30].

### 3.2. Fourier Transform Infrared (FTIR) of Melittin-Loaded Fibers

FTIR spectra were used to assess the level of drug–polymer interactions that represent the compatibility of melittin and PVP within the fibers as an indicator of high quality and stable composites that lack solid-phase separation [37]. The chemical structure of melittin is presented in Figure 1. Melittin is a polypeptide composed of 26 primary amino acids with a molecular formula of C_131_H_229_N_39_O_31_ [38]. In Figure 3, PVP showed bands between 3650 to 3050 cm^−1^ from (O-H stretching) due to the hydrophilicity and water adsorption properties of this polymer, in addition to 2840–3010 cm^−1^ (C-H stretches), 1660 cm^−1^ (C=O) and 1290 cm^−1^ (C-N stretch). Melittin demonstrated a broadband at 3400–3300 cm^−1^ of (NH_2_ stretching) in guanidine and amidic amino groups. There is also a broadband at 1700–1600 cm^−1^ tailored in the peptide backbone CO-NH region and at 1600–1500 cm^−1^, representing the N-H bending vibration of NH_2_, and C–O stretching vibrations from the C-terminal amino acid assigned at 1100–1250 cm^−1^ [39]. The spectra for the melittin-loaded fibers mat displayed notable differences from the raw melittin. These are most noticeable in the spectra at the stretches at 1500 cm^−1^, and 1620 cm^−1^ have both merged into the PVP C=O peak at 1700 cm^−1^ (Figure 3). In contrast, these changes are not noticed in the spectra of the physical mixture (PM, i.e., the raw mix of PVP and melittin at a similar polymer:drug ratio of the fibrous system). This shift in peak positions could indicate the molecular interaction between melittin and PVP, as explained in [40,41]. In addition, the presence of a distinctive peak of melittin (≈948 cm^−1^) in the melittin-loaded fibers, and the PM, suggests the presence of melittin in this fibrous system [42].

### 3.3. Drug Loading (DL) and Entrapment Efficiency (EE%) of the Melittin-Loaded Fibers Determination

The DL and EE% of melittin fibers were determined as 49 ± 3 μg/mg and 84 ± 5%, respectively, using the developed melittin UHPLC–MS/MS calibration curve illustrated in the Appendix A. Although having an EE% of >80% is considered high, the EE% of melittin-loaded fibers was slightly lower than previous PVP fibers studies that reported an EE% of >90% [43,44].

### 3.4. Drug Release of the Melittin-Loaded Fibers

The melittin-loaded fibers release study showed an initial burst release of approximately 40% after 5 min, then an 80% drug release at 30 min, and finally, a complete release after 180 min. This rapid release rate of melittin was due to the hydrophilic nature of the PVP polymer that accelerated the disintegration and dissolving of the fibers and the high surface-to-volume ratio of the fibers, which increased the contact area with the dissolution medium as presented in Figure 4. This release behavior was in agreement with Aburayan et al. shown with halicin/PVP fibers [30]; Li et al. shown with carvedilol/PVP fibers [45]; and Sriyanti et al. shown with α-mangostin/fibers [46].

### 3.5. In Vitro Cytotoxicity Assessment of Melittin

The in vitro cytotoxicity assessment of melittin is essential to evaluate its safety profile before being applied in the biomedical field. In this work, an MTS assay was conducted to assess the effect of several concentrations of melittin on the metabolic activity of HFF-1 following 24-, 48- and 72-h cell exposure. However, only the 24-h cell exposure was considered since the intended application of this fibrous system will not exceed 24 h. Figure 5 demonstrates high cell viability (%) of human cells with low concentrations of melittin detected. The viability was reduced to less than 20% at higher concentrations (20, 40 and 80 µg/mL). Increasing melittin’s HFF-1 cells exposure time to 48- and 72 h exhibited high cell viability only at the lowest concentration applied (0.625 µg/mL). In contrast, the relative cell viability of HFF-1 was reduced with concentrations of ≥1.25 µg/mL. The results revealed that the exposure time of the applied peptide has a significant impact on cellular metabolic activity. Therefore, a melittin concentration of ≤10 µg/mL is considered safe for use on human dermal fibroblasts HFF-1 after 24-h cell exposure but not after 48 or 72 h. Additionally, the cell line type may have an essential role in cell viability. A study by Askari et al. reported that the cell viability of the human primary fibroblast cell line (C654) following a 24-h cell exposure of melittin peptide was affected at a concentration of ≥2.5 µg/mL [47].

### 3.6. Minimum Inhibitory Concentration (MIC) of Melittin

Melittin demonstrated a concentration-dependent antimicrobial effect against different bacterial strains, in addition to *C. albicans* yeast. The MIC was determined in a concentration range of 0.01 and 20 µg/mL, and the MICs of melittin against all test microorganisms are shown in Table 1. Melittin MIC against a representative bacterium strain *S. aureus* ATCC 977 is demonstrated in the Appendix A. According to the cytotoxicity study of melittin in human dermal fibroblasts HFF-1, a ≤10 µg/mL concentration is considered safe on this cell line. Therefore, melittin, with an MIC of 5 μg/mL, is safe to be used for skin infections that are caused by the antibacterial-sensitive and -resistant strains of *S. aureus* and *A. baumannii* in addition to the antimicrobial-sensitive strain of *E. coli* and *C. albicans* yeast.

### 3.7. Zone of Inhibition Assessment of Melittin-Loaded Fibers

The antimicrobial activity of melittin fibers was assessed by the zone of inhibition assay against all bacterial and yeast strains tested in the MIC study. Well-defined zones of no growth (i.e., zones of inhibition) of melittin fibers and discs, as control, were detected with variable diameters, as shown in Figure 6. The zone of inhibition of melittin against a representative bacterium strain *S. aureus* ATCC 977 is demonstrated in the Appendix A. In equivalent loading amounts to melittin fibers, the inhibition zones for melittin discs were approximately similar. However, slight variations in the diameter against each microorganism have occurred, which could be due to the deviation in the formulation drug loading upon electrospinning or a processing error, such as the fibers’ inaccurate weighing that could lead to result variation. The essential outcome of this assay is that the melittin-loaded fibers have demonstrated antimicrobial activities against different bacterial and yeast strains that usually infect the human skin, and melittin has retained its antimicrobial effect after being formulated as electrospun fibers.

Over the last several years, electrospun fiber dressings have been widely investigated as a potential technique for preparing antimicrobial wound dressing. Moreover, they provide several essential advantages that make them suitable for use in wound dressings, including a physical barrier against bacteria invasion and biofilm development while still retaining oxygen transfer properties critical to the healing process [48]. In a similar study carried out by Khosravimelal et al., it was shown that an AMP-loaded electrospun bilayer scaffold could be used to treat wounds affected by MDR bacterial strains, in which a bilayered fibrous system loaded with 32 µg/mL of the CM11 peptide has an antibacterial activity without causing cytotoxicity [49]. In addition, scaffolds incorporating BV produced by Abou Zekry et al. displayed a slight increase in the activity against *S. aureus* but not against *E. coli* [50]. According to earlier studies, BV is more potent against gram-positive bacteria than gram-negative in damaging the outer membrane integrity [51]. Green nanofibrous dressings were also developed by Sarhan et al. and tested for antimicrobial and wound healing properties. The results showed that the BV-loaded nanofibers have effective antibacterial activities against MRSA and MDR *P. aeruginosa*, as well as wound healing profiles and good biocompatibility [52]. All these studies demonstrated the potential use of AMP, particularly BV, as efficient and safe antibacterial wound dressings against MDR bacterial strains.

## 4. Conclusions

The findings provided in this study will serve as a foundation for future peptide- nanofibrous scaffolds in biomedical applications. This study created a fibrous material system from a biocompatible polymer that could be used as a skin patch to treat skin infections caused by AMR bacteria. An active compound of bee venom, melittin, was successfully prepared and characterized using electrospinning technology. The SEM image showed that melittin-loaded fibers had smooth surfaces with no signs of beads or pores, indicating this fibrous system’s successful preparation. The average diameter of these fibers was measured to be 1030 ± 160 nm. The DL and EE% were 49 ± 3 μg/mg and 84 ± 5%, respectively. This high EE% can be considered another indicator of this system’s successful preparation. The in vitro release study of melittin-loaded fibers demonstrated that approximately 40% of melittin was released after 5 min, while a full drug release was achieved after 120 min. This release behavior was expected due to the rapid disintegration and dissolving of the PVP fibers. The in vitro cytotoxicity study of melittin showed that a concentration of ≤10 µg/mL is considered safe to use after a 24-h exposure on HFF-1 but not after 48 or 72 h. The antimicrobial MIC study exhibited an MIC of 5 μg/mL against antibacterial-sensitive and -resistant strains of *S. aureus* and *A. baumannii,* in addition to the antimicrobial-sensitive strain of *E. coli* and *C. albicans* yeast. The antimicrobial inhibition zone test of melittin fibers demonstrated the antimicrobial effectiveness of this system against all tested microbes, suggesting the potential use of a melittin-loaded fibrous scaffold as wound dressing patches for human skin infections. However, an in vivo assessment on an infected skin animal model should be considered to assess the safety and activity of this fibrous system before its clinical application.

## Figures and Tables

**Figure 1 pharmaceutics-14-00725-f001:**
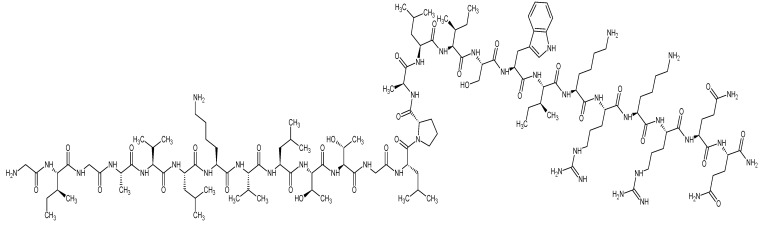
Melittin’s chemical structure which was drawn by ACD/ChemSketch.

**Figure 2 pharmaceutics-14-00725-f002:**
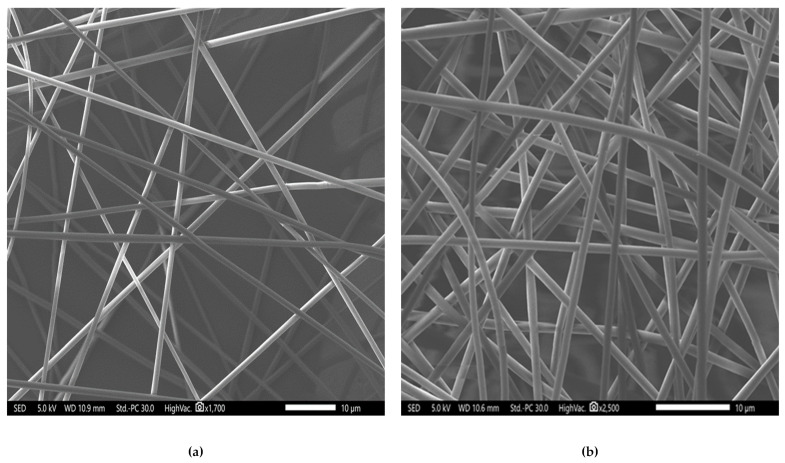
SEM images of blank (**a**) and melittin-loaded (**b**) fibers showed that both fibrous systems had smooth surfaces with no signs of bead or pores, with average diameters of 990 ± 130 nm, 1030 ± 160 nm, respectively.

**Figure 3 pharmaceutics-14-00725-f003:**
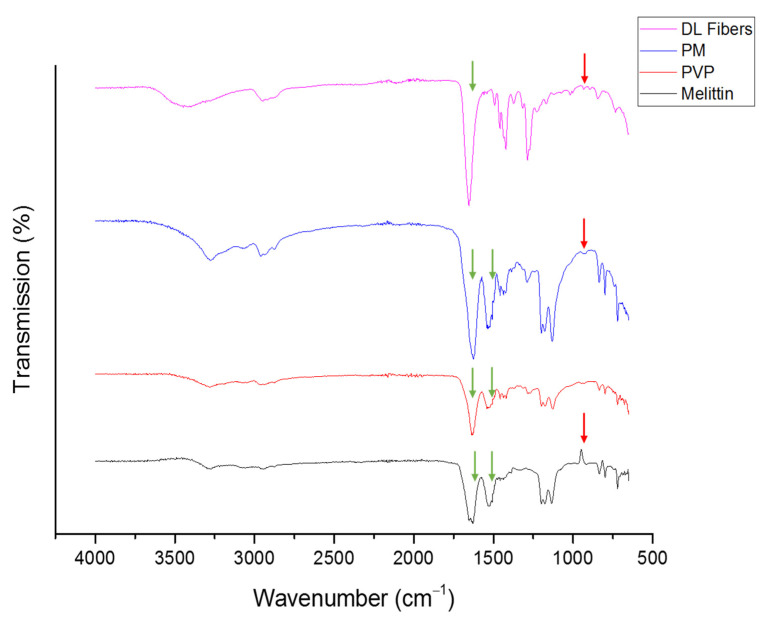
FTIR transmissions of PVP, melittin, PM and melittin fibers showing the characteristic peaks merge at 1500 cm^−1^ and 1620 cm^−1^ in the fibrous system but not in the PM (green arrows), which indicates the intermolecular formation bonds between melittin and PVP. A distinctive peak of melittin at 948 cm^−1^ which appeared in the PM and drug-loaded fibers and not in the PVP transmission suggests melittin presence in the fibrous system (red arrows). PM: physical mixture, DL: drug-loaded.

**Figure 4 pharmaceutics-14-00725-f004:**
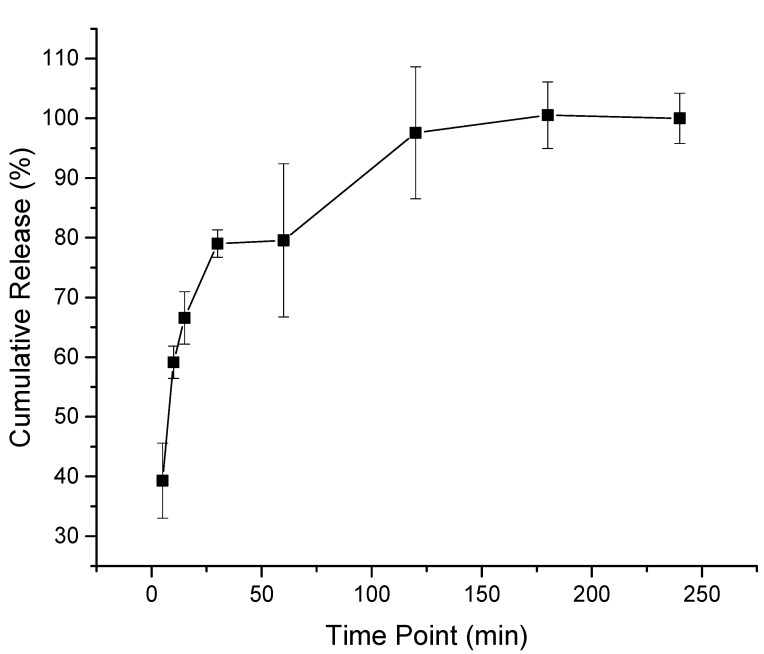
Melittin-loaded fibers release in PBS (pH 5.5), showing an initial burst release of 40% after 5 min and a complete drug release after 180 min. The results are presented as average ± SD of independent triplicates.

**Figure 5 pharmaceutics-14-00725-f005:**
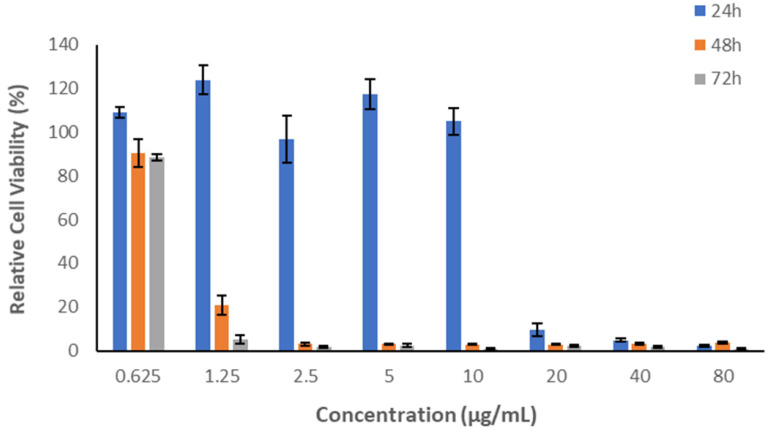
Cell viability of melittin at different concentrations exposed to HFF1 cells for 24, 48 and 72 h. The results show that melittin with a concentration of ≤10 µg/mL is considered safe after 24-h cell exposure but not after 48 or 72 h. The data of this MTS assay represented cell viability %, and the results are presented as average ± SD of independent triplicates.

**Figure 6 pharmaceutics-14-00725-f006:**
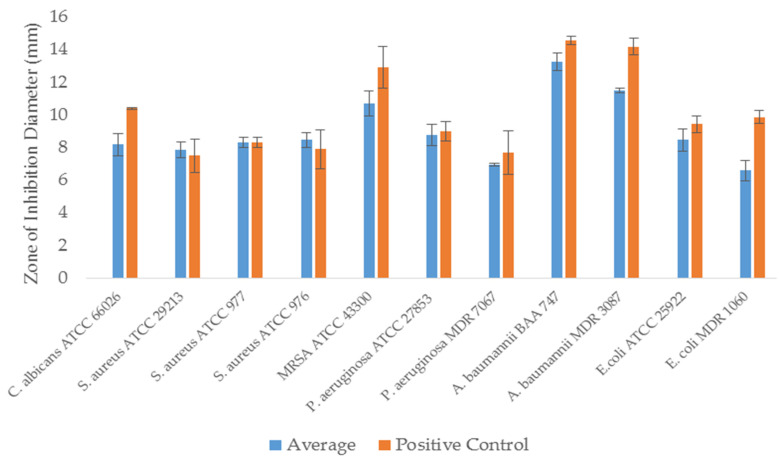
The zone of inhibition of melittin-loaded fibers against different antimicrobial-sensitive and -resistant bacterial strains, in addition to *C. albicans* yeast, compared to melittin containing discs (positive control). It shows that both melittin-loaded fibers and discs were effective against all strains when applied at equivalent amounts, suggesting that melittin retained its antimicrobial activity after electrospinning. The results are presented as average ± SD of independent triplicates.

**Table 1 pharmaceutics-14-00725-t001:** Melittin MIC against different antimicrobial sensitive and resistant bacterial strains of gram-negative and -positive, in addition to *C. albicans* yeast.

Microorganism	Melittin MIC (μg/mL)
*E. coli* ATCC 25922	5
*E. coli* MDR 1060	20
*A. baumannii* ATCC BAA 747	5
*A. baumannii* MDR 3087	2.5
*P. aeruginosa* ATCC 27853	20
*P. aeruginosa* MDR 7067	20
*S. aureus* ATCC 29213	5
*S. aureus* ATCC 976	5
*S. aureus* ATCC 977	5
*MRSA* ATCC 43300	5
*C. albicans* ATCC 66027	5

## Data Availability

The authors confirm that the data supporting the findings of this study are available within the article.

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
