# Peer review of "Melittin from Bee Venom Encapsulating Electrospun Fibers as a Potential Antimicrobial Wound Dressing Patches for Skin Infections"

_pharmaceutics, 2022, doi:10.3390/pharmaceutics14040725_

Round 1

Reviewer 1 Report

The authors have answered almost all queries. The quality of images has been improved. The in vitro experiment has been explained in detail. Although an in vivo experiment has not been added, the manuscript still looks better and completes the story. I recommend this manuscript for publication. 

Author Response

We thank the referee for their thoughtful and helpful comments.  

Reviewer 2 Report

After the revisions the quality of this manuscript has improved and can now be accepted.

Author Response

We thank the referee for their thoughtful and helpful comments.  

This manuscript is a resubmission of an earlier submission. The following is a list of the peer review reports and author responses from that submission.

Round 1

Reviewer 1 Report

The manuscript is generally well-written and the results are quite clearly presented. The conclusion is reasonably supported by the results. Nevertheless, the quality of the results should still be improved before acceptance.

  1. Figure 2. Pls add SEM image of fibers before or without melittin loading.
  2. Figure 6. The authors should show the toxicity on more than 1 cell type, e.g. include another endothelial cell, and longer duration, e.g. 48 or 72 hrs. It is too premature to conclude that melittin at a dose of ≤ 10 μg/mL is considered to be safe. Longer incubation time and effect on other healthy cells has to be investigated.
  3. Figure 7. Pls improve the graph quality. The chart is too small and error bars cannot be seen.
  4. Figure 8. Pls include statistical comparison between sample and control. In addition, please provide actual images showing the zone of inhibition.

Reviewer 2 Report

File attached
